# Feasibility of Non-Exposure Simple Suturing Endoscopic Full-Thickness Resection in Comparison with Laparoscopic Endoscopic Cooperative Surgery for Gastric Subepithelial Tumors: Results of Two Independent Prospective Trials

**DOI:** 10.3390/cancers13081858

**Published:** 2021-04-13

**Authors:** Bang Wool Eom, Chan Gyoo Kim, Myeong-Cherl Kook, Hong Man Yoon, Keun Won Ryu, Young-Woo Kim, Ji Yoon Rho, Young-Il Kim, Jong Yeul Lee, Il Ju Choi

**Affiliations:** Center for Gastric Cancer, National Cancer Center, 323 Ilsan-ro, Goyang 10408, Korea; kneeling79@ncc.re.kr (B.W.E.); mckook@ncc.re.kr (M.-C.K.); red1000@ncc.re.kr (H.M.Y.); docryu@ncc.re.kr (K.W.R.); gskim@ncc.re.kr (Y.-W.K.); coverstroy1@ncc.re.kr (J.Y.R.); 11996@ncc.re.kr (Y.-I.K.); jylee@ncc.re.kr (J.Y.L.); cij1224@ncc.re.kr (I.J.C.)

**Keywords:** laparoscopic and endoscopic cooperative surgery, endoscopic full-thickness resection, nonexposure technique, early gastric cancer

## Abstract

**Simple Summary:**

Nonexposure simple suturing endoscopic full-thickness resection (NESS-EFTR) is a recently developed method to prevent the exposure of tumor cells, and we performed a small prospective trial of NESS-EFTR for gastric subepithelial tumors (SETs). In this study, we compared the results of NESS-EFTR trial with those of another small prospective trial of laparoscopic and endoscopic cooperative surgery which was performed in different time period. The results of this study show the feasibility of NESS-EFTR for gastric SETs and provide evidence for the clinical application of the NESS-EFTR procedure.

**Abstract:**

Recently, nonexposure simple suturing endoscopic full-thickness resection (NESS-EFTR) method was developed to avoid tumor exposure to the peritoneal cavity. The aim of this study is to compare the short-term outcomes of the NESS-EFTR method with those of laparoscopic and endoscopic cooperative surgery (LECS) for gastric subepithelial tumors (SETs). A prospective single-center trial of LECS for gastric SETs was performed from March 2012 to October 2013 with a separate prospective trial of NESS-EFTR performed from August 2015 to June 2017, enrolling 15 patients each. Among the 30 enrolled patients, 14 who underwent LECS and 11 who underwent NESS-EFTR were finally included in the analysis. The rate of complete resection and successful closure was 100% in both groups. The operating time was longer for NESS-EFTR group than for LECS (110 vs. 189 min; *p* < 0.0001). There were no postoperative complications except one case of transient fever in the NESS-EFTR group. One patient in the LECS group had peritoneal seeding of gastrointestinal stromal tumor at 17 months postoperatively, and there was no other recurrence. Although NESS-EFTR had long operating and procedure times, it was feasible for patients with gastric SETs requiring a nonexposure technique.

## 1. Introduction

The standard surgical treatment for gastric subepithelial tumors (SETs) is local excision with a negative surgical margin [1,2,3]. Recently, laparoscopic gastric wedge resection has been widely performed for gastric SETs with the benefit of being a minimally invasive approach [4,5,6]. However, laparoscopic gastric wedge resection is performed outside the stomach, and in this extragastric approach, it is difficult to determine the appropriate resection line. Excessive resection can cause postoperative gastric deformity, followed by gastric stasis. In particular, lesions located near the esophagogastric junction or pyloric ring are technically challenging to resect using a laparoscopic approach. In these cases, total or distal gastrectomy is performed to avoid postoperative stenosis [7,8,9].

Laparoscopic and endoscopic cooperative surgery (LECS) was developed to overcome the limitations of laparoscopic wedge resection [10,11,12]. A circumferential incision is made using endoscopic submucosal dissection (ESD) devices and techniques, followed by seromuscular resection and closure using laparoscopic devices and techniques. LECS enables the determination of an appropriate surgical margin both vertically and laterally. However, an artificial perforation occurs during LECS, which can increase the risk of bacterial contamination and the dissemination of tumor cells to the peritoneum. Moreover, gastrointestinal stromal tumors (GISTs) with ulceration have a high risk of peritoneal dissemination [13].

We previously developed nonexposure simple suturing endoscopic full-thickness resection (NESS-EFTR) to prevent the spillage of the gastric contents into the peritoneum and the exposure of tumor cells. This technique includes laparoscopic seromuscular suturing, which results in the inversion of the stomach wall, endoscopic full-thickness resection (EFTR) of the inverted stomach wall, and endoscopic mucosal suturing using endoloops and clips. Complete resection and successful closure were achieved in previous NESS-EFTR procedures performed in a porcine model, and the safety and feasibility of NESS-EFTR were demonstrated [14,15].

We performed two independent prospective trials of LECS and NESS-EFTR in different time periods. A total of 15 patients were enrolled in each study, and technical safety and feasibility were evaluated. The aim of this study was to compare the short-term outcomes between the LECS and NESS-EFTR procedures for gastric SETs.

## 2. Materials and Methods

### 2.1. Study Design and Patients

An investigator-initiated, prospective, single-center trial of LECS for gastric SET was performed from March 2012 to October 2013. A separate prospective trial of NESS-EFTR was performed at the same institution from August 2015 to June 2017. A total of 15 consecutive patients with gastric SETs were enrolled in each trial, and the inclusion and exclusion criteria were as follows in both studies.

Inclusion criteria:Aged 20 years or older;Gastric SET that had invaded the muscularis propria on endoscopic ultrasound (EUS);Estimated tumor sizes of 1.5–5 cm (diameter) or an increase in tumor size during follow up;Agreement to participate in the clinical study through informed consent.

Exclusion criteria:Suspicious lymph node metastasis or tumor invasion to adjacent organs on preoperative EUS or computed tomography (CT);Inappropriate physical condition for surgery with general anesthesia;Presence of bleeding tendency.

All patients provided written informed consent before enrollment. The study protocol was approved by the local ethics committee of the National Cancer Center, Korea (Protocol Numbers NCCCTS12-604 and NCC-2015-0171 for the first and second trials, respectively), and the studies were registered at ClinicalTrials.gov (identifiers: NCT02042079 and NCT02764944, respectively).

### 2.2. LECS Procedure

The patient was placed in supine position under general anesthesia, and one to five ports were inserted (5 -mm or 12-mm ports) in the abdomen. In the single-port surgery, a four-hole single port (Octoport, Dalim, Wonju, Korea) was inserted in the umbilicus. In the multiport surgery, three to five ports were inserted in the umbilicus and the right upper, right lower, left upper, and left lower quadrants. After clamping of the proximal jejunum at 5 to 10 cm on the distal side of the ligament of Treiz, the tumor location was identified by endoscopy. An endoscopic circumferential mucosal incision was made around the lesion using an ITknife (precutting). After half the circumference was precut, a small puncture in the gastric wall was made on the mucosal incision line with a fixed flexible snare (Kachu Technology, Seoul, Korea). Endoscopic full-thickness resection (EFTR) was performed using an ITknife2 (Olympus, Tokyo, Japan) from the small puncture hole along the precut mucosal incision line until one-third or half of the circumference was resected. Additional laparoscopic full-thickness resection using an ultrasonically activated device (Harmonic Scalpel, Ethicon Endo-Surgery, Cincinnati, OH, USA) was permitted if needed. After the partially resected SET was exposed to the peritoneum and lifted by laparoscopic forceps, laparoscopic linear stapling devices (Echelon 60, Ethicon Endo-Surgery) were used for complete resection and suturing (Figure 1). Resected tumor tissue was retrieved via umbilical incision using an extraction bag.

### 2.3. NESS-EFTR Procedure

A detailed description of NESS-EFTR is provided in our previous report on the animal model [14,15]. After insertion of two to five ports in the abdomen, endoscopic circumferential incision of the mucosal layer around the lesion was performed. When the tumor boundary was clearly visible, only saline was injected around the tumor (Figure 2). Laparoscopic serosal marking was then performed on the opposite side (serosal surface) of the endoscopic mucosal incision line (mucosal surface). When an endoscopist pressed the mucosal incision line with the tip of a fixed flexible snare inside the stomach, a surgeon marked the pressed spots using a monopolar device outside of the stomach. Laparoscopic seromuscular suturing was performed via a continuous method using unidirectional barbed thread (V-loc 180 3-0, Covidien, Mansfield, MA, USA). An interrupted method using black silk thread (MERSILK 3-0, Ethicon, Somerville, NJ, USA) was also allowed according to the surgeon’s preference. This procedure resulted in inversion of the stomach wall. EFTR of the inverted stomach wall was performed from inside the stomach using an ITknife2 or a conventional needle knife (Needle Papillotome, MTW Endoskopie, Wesel, Germany). Resected tissues were grasped endoscopically with alligator jaws (FG-6L-1, Olympus) and retrieved via the oral cavity. Finally, endoscopic mucosal suturing with endoloops was performed. An open endoloop with a 30 mm diameter (MAJ-340, Olympus, Tokyo, Japan) was positioned along both edges of the resection site with three to five clips (HX-610-90L or HX-610-135L, Olympus, Tokyo, Japan). The endoloop was then closed and released. Two or three endoloops were required for complete mucosal closure of the resected site.

### 2.4. Follow Up Surveillance

When patients were diagnosed with benign tumors in the final pathological report, esophagogastroduodenoscopy was performed 3 months after surgery to monitor the anastomosis site, and annual endoscopic evaluation was recommended. For patients diagnosed with GIST with moderate or high risk, adjuvant treatment with imatinib was recommended and most patients received adjuvant chemotherapy [16]. After completion of adjuvant chemotherapy, short-term follow up at 3–6 months with computed tomography was performed.

### 2.5. Outcome Measurements

The primary outcome was the rate of complete resection, which was defined as successful en bloc resection (resected tumor in one piece) with a clear margin. Secondary outcomes were the rate of successful closure, procedure times, and anastomosis-related complications such as leakage or stenosis, which were assessed according to the Clavien–Dindo classification [17]. Risk stratification of GIST was performed according to the 2010 National Comprehensive Cancer Network guidelines [18].

### 2.6. Statistical Analysis

Continuous values are presented as the mean ± standard deviation or median with interquartile range (IQR), and categorical variables are shown as proportions. Distribution differences were tested using the Wilcoxon rank sum test or t test for continuous variables and the chi square test or the Fisher exact tests for categorical variables. Statistical analyses were performed using SAS version 9 software (SAS Institute, Cary, NC, USA). *p* values less than 0.05 were considered to indicate statistical significance.

## 3. Results

### 3.1. Clinicopathological Characteristics

In the LECS trial, one patient was excluded because the tumor was identified as an extragastric mass during the operation. In the NESS-EFTR trial, three cases (one hepatic hemangioma and two large exophytic tumors that were confirmed during the operation) were excluded as screening failure and one patient withdrew from the study. In total, 14 and 11 patients were included in the analysis.

The clinicopathological characteristics of the patients in the two trials are shown in Table 1. The median age was 61 and 56 years, and the male proportion was 21.4% and 45.5% in the LECS and NESS-EFTR groups, respectively. The majority of tumors were located in the cardia/fundus (64.3% (9/14) and 72.7% (8/11), respectively), and the distance between the tumor and the esophagogastric junction was less than 5 cm. The most common pathology diagnosis was GIST in both groups.

### 3.2. Procedure Times

Table 2 shows the overall operating and procedure times in both groups. Tumor localization involved endoscopic mucosal marking in the LECS group and endoscopic mucosal marking plus laparoscopic serosal marking in the NESS-EFTR group. The laparoscopic serosal marking can be achieved by an endoscopist’s assistant, who is pushing the mucosal edge with an endoscopic tool. Therefore, the time for the surgeon and the endoscopist to cooperate was included in the tumor localization time in the NESS-EFTR group. Suturing involved laparoscopic stapling in the LECS group and laparoscopic serosal simple suturing plus endoscopic suturing using endoloops and clips in the NESS-EFTR group. The NESS-EFTR group had longer operating times than the LECS group (189 vs. 110 min, *p* < 0.001). The time for each procedure, including tumor localization, EFTR, and suturing, was also longer in the NESS-EFTR group.

### 3.3. Surgical Outcomes

The rates of complete resection and successful closure were 100% in both trials (Table 3). Intraoperative perforation developed in two patients (18.2%) during NESS-EFTR, and laparoscopic reinforcement suturing was performed for the perforation site. There were no other intraoperative complications such as bleeding requiring transfusion or adjacent organ injury.

In the postoperative period, there was one event in the NESS-EFTR group. A patient who had laparoscopic reinforcement suturing due to perforation during NESS-EFTR had no specific symptoms until postoperative day four. On postoperative day five, follow up esophago-gastro-duodenoscopy was done, and no leakage or stenosis was observed. However, flatulence and fever developed after the endoscopic evaluation, and there was a small amount of pneumoperitoneum in the lesser sac and the perihepatic space as evidenced by computed tomography. Empirical antibiotics were administered (Grade II of the Clavien–Dindo classification), and the fever subsided. There were no other complications in either group. The time to the start of oral intake and the length of the hospital stay were not significantly different between the groups.

### 3.4. Follow Up Results

In the endoscopic evaluation three months after surgery, there were no abnormal findings such as delayed leakage, stenosis, or food stasis in either group.

The median follow up period was 72 months (IQR, 18.3–77.3 months) and 28 months (IQR, 15.0–35.0 months) in the LECS and NESS-EFTR groups, respectively. In the LECS group, six patients were diagnosed with GIST with moderate or high risk and three patients received adjuvant treatment with imatinib. One patient with a moderate-risk GIST (2.5 cm, mitotic count: 27/50 high-power field) refused adjuvant therapy because the government health insurance system did not cover the adjuvant treatment for moderate-risk GIST. This patient experienced recurrence with multiple peritoneal seeding 16 months after LECS and received palliative imatinib therapy for five years.

In the NESS-EFTR group, five patients were diagnosed with GIST with very low or low risk and one patient had a moderate-risk GIST. The patient with a moderate-risk GIST received adjuvant imatinib treatment, and no recurrence was seen until during the 54 months of follow up.

## 4. Discussion

This study is the first report of NESS-EFTR for patients with gastric SETs, and the surgical outcomes of NESS-EFTR were compared with those of LECS. We achieved complete resection and successful closure rates of 100% with both procedures, and there were no severe postoperative complications in either group. Although NESS-EFTR had a longer operating time than LECS due to additional procedures for nonexposure and hard-to-access tumor location (cardia/fundus), it is feasible for gastric SETs and has the advantage of no peritoneal contamination.

The LECS procedure, which combines laparoscopic gastric resection with ESD, was developed in 2008 [10]. The main advantage of this procedure is its ability to determine appropriate resection lines via the confirmation of the exact tumor localization endoscopically. Many studies have demonstrated the feasibility of LECS for gastric SETs [19,20,21], but the LECS procedure caries a risk of gastric content spillage and peritoneal seeding of cancer cells due to an intentional gastric perforation. Cancer cells in early gastric cancer are easily detached via contact with the tumor surface (27.6%) [22]. In the present study, one patient who underwent LECS for gastric GIST experienced peritoneal recurrence.

After completion of the prospective LECS trial, we developed a nonexposed technique to overcome the shortcomings of LECS, which is NESS-EFTR. Two feasibility studies were performed in a porcine model to confirm the safety and technical feasibility of the NESS-EFTR [14,15]. In these animal studies, complete resection was achieved in all pigs undergoing the procedure, and no early death due to complications was observed. The prospective trial of NESS-EFTR for the patients with gastric SET was the next step. Despite technical difficulties due to tumor location (the tumors were mostly located in the cardia/fundus), complete resection was achieved in all NESS-EFTR cases without severe postoperative complications.

NESS-EFTR is a more complicated procedure and required longer operating times in this study. In LECS, the perforation site conducted by an endoscopist can be easily detected by a surgeon; a serosal marking is not necessary. However, in NESS-EFTR, serosal marking was additionally performed for tumor localization based on close cooperation between a surgeon and an endoscopist, which takes considerable time. In particular, most cases were intraluminal cardiac tumors, which are difficult to approach both endoscopically and laparoscopically. In the LECS group, EFTR was performed over one-third or half of the circumference, with the remaining half resected using laparoscopic stapling; in contrast, the whole circumference of the tumor was resected endoscopically in the NESS-EFTR group. Endoscopic mucosal suturing with snaring and clips was also added to the NESS-EFTR procedure. For these reasons, NESS-EFTR had a longer operating time than LECS.

Despite its longer operating time, NESS-EFTR has the critical advantage of being a nonexposure technique. NESS-EFTR might be oncologically safe for cases of SET with surface ulcers. The operating time could be also decreased in the future as surgeons and endoscopists gain experience with the technical problems and new instruments are developed. Another advantage of the NESS-EFTR is that no stapler is used, which can be financially beneficial in the countries where laparoscopic staplers are not covered by the public health insurance system.

NESS-EFTR is similar to nonexposed endoscopic wall-inversion surgery (NEWS), in such a way that a bowel wall is inverted [23,24,25,26]. However, there are some theological advantages in NESS-EFTR compared with NEWS. Although gastric tumors are located on the mucosa, circumferential incision around the tumor is laparoscopically performed at the serosal side in NEWS, and the seromuscular incision line can be a little different from the actual tumor border at the mucosal side. Conversely, circumferential mucosal incision around the tumor is conducted on the mucosal side in the NESS-EFTR procedure, which can be more helpful in obtaining a safe margin from the tumor. In addition, circumferential mucosal incision around the tumor in NESS-EFTR is the same procedure as endoscopic submucosal dissection and can be quickly and easily performed by an endoscopist. NESS-EFTR has another advantage compared with NEWS in that endoloops and endoscopic clips are used for endoscopic suture on the mucosal side. The suture of the mucosal side is not mandatory, and only clips are used if the suture of the mucosal side is tried in NEWS. The use of endoscopic clips is still a widely accepted closure technique, but the effectiveness of the metallic clips is more dependent on the endoscopists’ skill. Endoloops with clips are used in NESS-EFTR, which provides more secure closure [27].

In combined laparoscopic and endoscopic approaches for neoplasia with the nonexposure technique (CLEAN-NET), the tumor is removed using a laparoscopic linear stapler after a laparoscopic seromuscular incision around the tumor [28,29,30]. Since the mucosal margin from the tumor cannot be identified during stapling in CLEAN-NET and a pathological examination is difficult for the stapler line, verifying whether safe margins are achieved or not may be unclear.

This study has several limitations. First, the purpose of each trial was to evaluate the technical feasibility of each procedure, and the sample size was small. A total of 14 and 11 patients were included in the analysis for the two groups. Second, the follow up period of the NESS-EFTR group was significantly shorter than that of the LECS group because the study period was different. Moreover, the follow up schedules differed according to the patients’ pathological diagnoses, and there was a large variation in the follow up period among patients. Finally, the procedure time was highly affected by tumor location and tumor size. Skillful endoscopic techniques and considerable experience are also required for successful NESS-EFTR.

## 5. Conclusions

Although NESS-EFTR had long operating and procedure times, it was feasible for patients with gastric SETs requiring a nonexposure technique. Further large-scale studies with long-term follow up are needed.

## Figures and Tables

**Figure 1 cancers-13-01858-f001:**
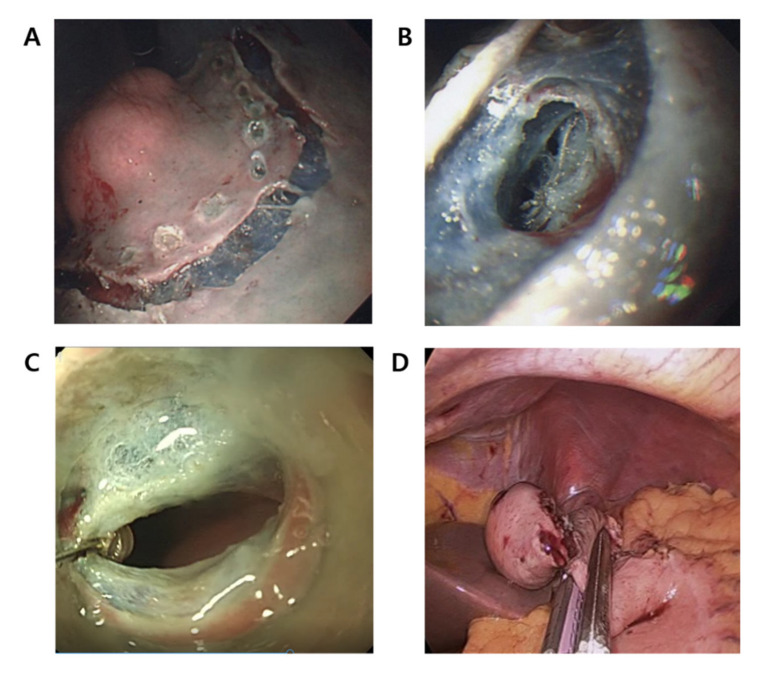
The laparoscopic and endoscopic cooperative surgery (LECS) procedures. (**A**) Endoscopic submucosal resection around the tumor. (**B**) Introduction of a small puncture in the gastric wall on the mucosal incision line with a fixed flexible snare. (**C**) Endoscopic full-thickness resection (EFTR) along the precut mucosal incision line until one-third or half of the circumference is resected. (**D**) Closure at the incision line using a laparoscopic stapling device.

**Figure 2 cancers-13-01858-f002:**
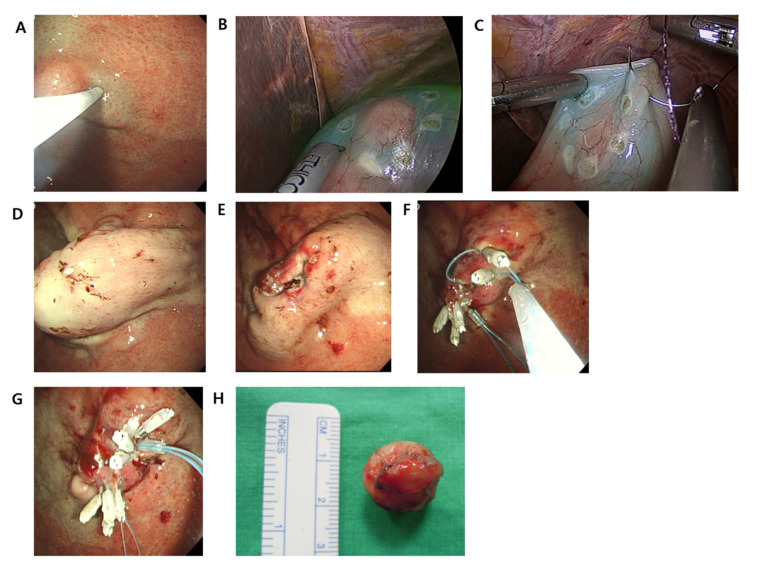
The nonexposure simple suturing endoscopic full-thickness resection (NESS-EFTR procedures). (**A**) Endoscopic saline injection around the tumor. (**B**) Laparoscopic serosal marking. (**C**) Laparoscopic seromuscular suturing. (**D**) Endoscopic view after serosal suturing. (**E**) Completion of endoscopic full-thickness resection. (**F**) Endoscopic suturing using endoloops and clips. (**G**) Completion of endoscopic suturing. (**H**) Resected specimen.

**Table 1 cancers-13-01858-t001:** Baseline characteristics.

Characteristics	Subgroup	LECS (*N* = 14)	NESS-EFTR (*N* = 11)	*p*-Value
Age (year) (median, IQR)		61.0 (51.0, 66.8)	56.0 (40.0, 59.0)	0.202
Sex (%)	Male	3 (21.4)	5 (45.5)	0.389
	Female	11 (78.6)	6 (54.5)	
BMI (kg/m^2^) (median, IQR)		24.2 (22.2, 25.6)	24.0 (22.8, 25.9)	0.661
Location of tumor	Body	5 (35.7)	3 (27.3)	0.999
	Fundus	2 (14.3)	2 (18.2)	
	Cardia	7 (50.0)	6 (54.5)	
Distance between the tumor and esophagogastric junction (cm)	≤2	7 (50.0)	6 (54.5)	0.999
	>2, and ≤5	2 (14.3)	2 (18.2)	
	>5	7 (50.0)	6 (54.5)	
Tumor size (cm) (median, IQR)		2.6 (2.3, 3.7)	2.2 (1.5, 3.0)	0.12
Pathological diagnosis	GIST	9 (64.3)	6 (54.5)	0.343
	Leiomyoma	3 (21.4)	5 (45.5)	
	Schwannoma	2 (14.3)	0 (0)	

LECS, Laparoscopy and endoscopic cooperative surgery; NESS-EFTR, nonexposure simple suturing endoscopic full-thickness resection; IQR, interquartile range; BMI, body mass index; GIST, gastrointestinal stromal tumor.

**Table 2 cancers-13-01858-t002:** Procedure times.

Procedure Times	Subgroup	LECS (*N* = 14)	NESS-EFTR (*N* = 11)	*p*-Value
Operating time (min) (median, IQR)		110.0 (96.3, 151.3)	189.0 (170.0, 230.0)	<0.001
Procedure time (min) (median, IQR)	Tumor localization	2.0 (1.0, 2.25)	18.0 (5.0, 37.0)	<0.001
	EFTR	13.0 (9.8, 21.3)	36.0 (19.0, 51.0)	<0.001
	Suturing	25.0 (13.5, 38.3)	50.0 (34.0, 64.0)	0.005

LECS, Laparoscopy and endoscopic cooperative surgery; NESS-EFTR, nonexposure simple suturing endoscopic full-thickness resection; IQR, interquartile range; EFTR, endoscopic full-thickness resection.

**Table 3 cancers-13-01858-t003:** Surgical outcomes.

Surgical Outcomes	Subgroup	LECS (*N* = 14)	NESS-EFTR (*N* = 11)	*p*-Value
Complete resection (*n*, %)		14 (100)	11 (100)	
Rate of successful closure (*n*, %)		14 (100)	11 (100)	
Conversion to open surgery (*n*, %)		0 (0)	0 (0)	
Postoperative complications (*n*, %)	Focal peritonitis	0 (0)	1 (9.1) *	
	Leakage/stenosis	0 (0)	0 (0)	
Time to start of oral intake (day) (median, IQR)		1.0 (1.0, 1.5)	1.0 (1.0, 2.0)	0.434
Hospital stay (day) (median, IQR)		5.0 (4.0, 5.5)	5.0 (5.0, 6.0)	0.12

LECS, Laparoscopy and endoscopic cooperative surgery; NESS-EFTR, nonexposure simple suturing endoscopic full-thickness resection; IQR, interquartile range. * One patient who had perforation during the NESS-EFTR had flatulence and fever after follow up esophago-gastro-duodenoscopy on postoperative day 5. No leakage or stenosis was observed in the endoscopic evaluation and there was small amount of pneumoperitoneum in the lesser sac and the perihepatic space as evidenced by computed tomography. Empirical antibiotics were administered and the fever subsided.

## Data Availability

The data presented in this study are available on request from the corresponding author. The data is not publicly available due to patient privacy and the General Data Protection Regulation.

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
