# Peer review of "Feasibility of Non-Exposure Simple Suturing Endoscopic Full-Thickness Resection in Comparison with Laparoscopic Endoscopic Cooperative Surgery for Gastric Subepithelial Tumors: Results of Two Independent Prospective Trials"

_cancers, 2021, doi:10.3390/cancers13081858_

Round 1
Reviewer 1 Report
The authors of the article "Feasibility of Non-Exposure Simple Suturing Endoscopic Full-thickness Resection in comparison with Laparoscopic Endoscopic Cooperative Surgery for Gastric Subepithelial Tumors: Results of two independent prospective trials" have promptly responded to the queries raised previously. I don't have any additional query or doubts pertaining to this article.
Author Response
Thank you for the kind comment for our manuscript.
Reviewer 2 Report
Dear authors,
The revised manuscript has addressed the comments previously made.
1) After going again through the procedure description there are some differences between the paragraph 2.3 NESS-EFTR procedure and Figure 2. In particular, in L115-117 it is stated that "endoscopic circumferential incision of the mucosal layer around the lesion was performed (figure 2). When the tumor boundary was clearly visible, only saline was injected around the tumor. However in figure 2 it is shown that the first step is the injection of saline around the tumor. Can the authors clarify what is the protocol for their procedure?
2. In L293-295 the authors added the following "Since the mucosal margin from the tumor cannot be identified during stapling in CLEAN-NET and a pathological examination is difficult for the stapler line, verifying whether safe margins are achieved or not may be unclear [18]". Reference no 18 is irrelevant and does not support this statement.
3. The added segments require some minor English language polishing.
Author Response
Reply for comment 1>
Thank you for your meticulous review and comments. We agree that readers can be confused about endoscopic mucosal incision because figure 2A shows just saline injection around the tumor. Actually, endoscopic circumferential incision is mandatory for early gastric cancers in our current clinical trial of NESS-EFTR. However, most subepithelial tumors had clearly visible tumor boundary and just saline injection was performed to them in this study. So we moved “(figure 2)” to the end of the sentence as follows.
Page 3
~ endoscopic circumferential incision of the mucosal layer around the lesion was performed (figure 2). When the tumor boundary was clearly visible, only saline was injected around the tumor (figure 2).
Reply for comment 2>
Thank you for your detailed comment. We agreed that reference no.18 does not support the statement and this is our mistake. Actually, we added reference no.18 to support the safety of endoloops with clips for the last statement of previous paragraph (You can check it in the first “point by point response” in which reference no. 18 was added the end of previous paragraph). Probably, “[18]” was pushed back when the paragraph of CLEAN-NET was inserted. We really apologize this clear mistake and appreciate your meticulous review again. We moved “[18]” to the end of previous paragraph as follows. Moreover, it is just authors’ (including a pathologist) opinion that pathological examination is difficult for the stapler line, and we have no reference for it.
Page 8
Endoloops with clips are used in NESS-EFTR, which provides more secure closure [18].
Reply for comment 3>
Thank you for your comment. Actually, this manuscript may has some shortcomings in English language. However, we received second English editing service from Editage just before re-submission. Unfortunately, we have no certificate because this is the second editing for some revised sentences. So we will accept any additional grammatical revisions from editorial department of “Cancers”.

This manuscript is a resubmission of an earlier submission. The following is a list of the peer review reports and author responses from that submission.
Round 1
Reviewer 1 Report
This is a very interesting paper comparing the clinical results of two techniques that can be used for the resection of gastric submucosal tumors. The paper compares the clinical outcomes of LECS versus NESS-EFTR which is a technique developed by the authors. NESS-EFTR is a modification of the Non-Exposure Wall Inversion Surgery technique (NEWS) first described by Goto et al in an animal model in 2011. The authors have first tested their technique in an animal model and in this paper they present their clinical results comparing them with the results they have had with LECS for resection of submucosal tumors. The paper is well written and deals with a very interesting object, but has some issues to be addressed:
1) The study is described as a trial but it does not fulfill the criteria of a clinical trial. It is a cohort study where two patient cohorts are compared (NESS-EFTR versus LECS). It is also not clear why there is such a large time gap between October 2013 and August 2015. If the authors prospectively collected data on their NESS-EFTR patients and then compared them with the historic data they had on LECS patients this should be clearly stated in the text.
2) There is no mention of the methods used for statistical testing. The statistical tests used (e.g. chi-square, t-test, ANOVA etc) should be stated in the statistical analysis paragraph.
3) In table 2, the title "Tumor localization" is ambiguous. In the discussion it is insinuated that in the NESS-EFTR group the tumor localization time was prolonged due to the need for serosal marking. However it is not clear if the "Tumor localization" time includes only the mucosal and serosal marking or also the time for tumor exposure by the endoscopist and laparoscopic surgeon. Please clarify this in the study methodology.
4) The tumor position is given as "Body", "Fundus", "Cardia" but there is no mention if it was on the anterior or the posterior gastric wall. In addition there are no data describing the distance from the gastroesophageal junction or the pylorus. This is important since tumors in proximity of these two structures are more challenging as the authors state in their introduction.
5) In the Surgical Outcomes, there is a postoperative complication vaguely described as: "one patient in the NESS-EFTR group had a fever on post-operative day 5. However, no leakage or stenosis was observed on esophagogastroduodenoscopy. Empirical antibiotics were administered (Grade II of the Clavien-Dindo classification), and the fever subsided." However in table 3 this patient is clearly classified as having local peritonitis and it is explained that he had abdominal pain, fever and CT scan findings suggestive of air leak from the gastrostomy. This information should be moved to the Surgical Outcomes section to make it clearer for the reader.
6) The last paragraph of the discussion is not related to the paper and should be removed.
7) Reference 15 is a review paper written by one of the authors, and not an original paper. It may be considered inappropriate self-citation by the journal.
8) It might be beneficial to cite the original paper by Goto et al describing the Non-Exposure Wall Inversion Surgery technique (NEWS) while mentioning the differences between it and the authors' technique of NESS-EFTR.
9) An important advantage of the NESS-EFTR technique vs LECS is the omission of a stapler for the gastrostomy closure. This makes the technique more difficult from the laparoscopic surgeon's point of view but significantly reduces the cost. In countries where these procedures are not covered by the public health insurance system this is a very important benefit.
10) In the data availability statement there is a small typographical error. It should be "The data is not publicly available".
11) What follow-up did the authors do for patients with moderate or high risk GIST after the short term follow-up at 3-6 months? Did they use endoscopy, CT scans or both?
Reviewer 2 Report
The authors investigated the feasibility and safety Non-Exposure Simple Suturing Endoscopic Full-thickness Resection (NESS-EFTR) for gastric subepithelial tumors (SETs) compare to Laparoscopic Endoscopic Cooperative Surgery. They developed NESS-EFTR as a novel endoscopic treatment for SETs, and they demonstrated the excellent result. The manuscript was clearly summarized. Although I read your paper with interest, the findings are limited by several important factors. The followings should be taken into consideration.
- Although it is not specified in the paper, is this study set up as a non randomized clinical trial?
This study should be a feasibility study because there is no set sample size and the treatments being compared are from different time periods. In that case, the title itself needs to be changed.
- This technique seems to be quite similar to the previously reported Non-exposed endoscopic wall-inversion surgery (NEWS), so it may not be very novel. Furthermore, what are the advantages of this technique compared to NEWS?
- Please describe about the advantages of this technique compared to other non-exposure techniques such as CLEAN-NET and NEWS.